Enhancement of small doppler frequencies detection for LFMCW radar

http://orcid.org/0000-0002-3579-1469 Ghanem Sameh samehghanem@eaeat.edu.eg
Electronics and Communications, Egyptian Academy for Engineering and Advanced Technology (EAEAT) , Cairo, Al Qahirah , Egypt
Al-Hadhrami Tawfik
Electronic publication date: 2021 Jan 28
Publication date: 2021
Volume: 7
Electronic Location ID: e367
Received 2020 Oct 27; Accepted 2020 Dec 31
Copyright: © 2021 Ghanem
Copyright year: 2021
Copyright holder: Ghanem
License: This is an open access article distributed under the terms of the Creative Commons Attribution License, which permits unrestricted use, distribution, reproduction and adaptation in any medium and for any purpose provided that it is properly attributed. For attribution, the original author(s), title, publication source (PeerJ Computer Science) and either DOI or URL of the article must be cited.
License URL: https://creativecommons.org/licenses/by/4.0/

Keywords: LFMCW radar, SDLC–MTI, Doppler frequency, 2D-FFT, Signal processing

Funding: The authors received no funding for this work.

==============================
Detection of targets with small Doppler frequencies of linear-frequency modulated continuous wave radars is the main task of this article. The moving target indicator (MTI) is used to reject the fixed targets and high-speed targets through the radar research area. In this work, targets with small Doppler frequencies can be detected perfectly based on the frequency response of a single delay line canceller followed by single delay line integrator. An enhancement of the proposed algorithm is achieved using a filter in the range direction of the range-Doppler processor scheme. The proposed filter is chosen with certain coefficients after the first fast Fourier transform processor in range to enhance the radar performance. The evaluation of the proposed algorithm is achieved at different slow Doppler scenarios of the target and compared with the traditional algorithm which uses only MTI processor. Another aspect that is important for evaluation of the proposed algorithm is the detection performance of the algorithms through the receiver operating characteristic curves. Implementation of the proposed algorithm using FPGA is performed in real time applications and it is found that it meets the simulation results.

Introduction

Detection of slow moving targets is an important for linear-frequency modulated continuous wave (LFMCW) radars based on traditional techniques such as fast Fourier transform (FFT) in both range and Doppler directions (Skolnik, 2008). Usage of FMCW radar due to many advantages such as its small weight, small energy consumption and less hardware complexity relative to other radars (Lee & Kim, 2010). The target information such as range and speed can be extracted from LFMCW radars using two-dimensional FFT algorithm. The moving target indicator (MTI) is used to distinguish between the fixed and moving targets. There are many researches that enhance the detection of LFMCW radars using different techniques. In Salem et al. (2015), target detection of LFMCW radars is enhanced using Compressive Sensing theory in Doppler direction. In Salem et al. (2016), the authors investigate the real time implementation of the proposed algorithm for LFMCW radar. An enhancement of target detection in both range and Doppler directions based on CS is shown in Hossiny et al. (2018). In Ahmed (2019), the author enhances the detection of slow Doppler frequencies based on frequency response of both the single delay line canceller (SDLC) and integrator. The authors in Winkler (2007), achievement of Range-Doppler detection of automotive FMCW radar is performed to extract the target information based on FFT calculations.

In this article, an enhancement of small Doppler target detection is achieved using a proposed filter in range direction of FFT processor. The evaluation of the proposed processor has performed using MATLAB simulation and receiver operating characteristic (ROC) curves. Implementation of the proposed processor is designed and tested using FPGA. The organization of this paper is achieved as follows; after the introduction, “LFMCW Radar Detection and Processing” introduces a review on LFMCW radar processing and detection. “The Proposed Processor” illustrates on the operation of the proposed processor. Experimental results using MATLAB is illustrated in “Computer Simulation”. “Hardware Implmentation” presents the hardware implementation of the proposed processor using FPGA. Finally, the conclusion comes in “Conclusion”.

Lfmcw radar detection and processing

The general block diagram of LFMCW radar is shown as in Fig. 1. It consists of a transmitter, a receiver, mixer, and Analog-to-Digital converter (A/D). The received radar signal is processed after digitization using A/D converter in the form of base band signal. The target decision is made using the constant false alarm rate (CFAR) algorithm after Range-Doppler processing based on FFT.

Figure 1 General block diagram of LFMCW radar.

The transmitted signal of an FMCW radar can be modulated as follow (Levanon & Mozeson, 2004): (1) ST(t)=ATcos(2πfct+2π∫0t⁡fT(τ)dτ)

Where fT(τ)=BT.τ is the linear transmitted frequency as function of time, fc is the carrier frequency, B is the bandwidth, AT is the transmitted signal amplitude, and T is the time duration.

The received signal after reflection with delay of td=2.Ro+vtC and Doppler shift offD=−2.fcvC, the received frequency can be expressed as: (2) fR(t)=BT(t−td)+fD

Where Ro is the initial target range and v is the target velocity.

The received radar signal can be expressed as: SR(t)=ARcos(2πfc(t−td)+2π∫0t⁡fR(τ)dτ)

(3) =ARcos(2πfc(t−td)+BT(12t2−td.t)+fD.t)

Where AR represents the received signal amplitude. The target information can be obtained by mixing the transmitted and received signals in time domain and filtered using low-pass filter (LPF) to generate the intermediate frequency (IF) signal SIF(t) as: (4) SIF(t)=12cos(2π(fc.2RoC)+2π(±2RoC.BT+2fcvC)t)

The sign ± represents up and down ramp respectively. Therefore, beat frequency (fb) can be obtained in the spectrum of the baseband signal as: (5) fb=±2RoC.BT+2fcvC

The relation between the beat frequency (fb) and range (R) for fixed target is given by Komarov & Smolskiy (2003) and Levanon & Mozeson (2004) (6) fb=2RfmΔFC

Where fm is the modulated frequency, Δf is the receiver bandwidth and C is speed of light. Extraction of target information such as range and speed based on 2D-FFT is illustrated as shown in Fig. 2.

Figure 2 LFMCW radar signal processing using 2D-FFT.

According to the traditional algorithm for LFMCW radar, the spectrum of received radar signal is processed using FFT in range direction followed by FFT in Doppler direction. The output of second FFT is applied to CFAR processor to make a decision for target detection. One of enhancement method for target detection using SDLC-MTI followed by integrator (Ahmed, 2019) is illustrated in Fig. 3.

Figure 3 Block diagram of SDLC.

Block diagram of SDLC/SDLI algorithm.

The frequency response of SDLC MTI is multiplied with that of Single Delay Line Integrator (SDLI) as shown in Fig. 4. Figure 5A represents the realization of stable SDLI and Fig. 5B illustrates its frequency response at different values of gain (A).

Figure 4 Single Delay Line Integrator Structure.

(A) Stable realization. (B) Frequency response at different values of A.

Figure 5 LFMCW radar processor based on SDLC.

LFMCW radar processor based on SDLC/SDLI processor.

This structure has a good performance for slowly targets with small Doppler frequencies but has a bad evaluation for middle Doppler targets. This problem has been enhanced in Ahmed (2019) but with combined structure of the traditional algorithm (MTI with 2D-FFT processor) and the SDLI with Doppler FFT as shown in Fig. 6. The problem of this combination is the complexity which uses extra Doppler FFT processor in addition to SDLI processor. This problem can be overcame using the proposed processor or filter instead of high complexity as discussed in the next section.

Figure 6 Block diagram of LFMCW radar with the combined structure.

The proposed processor

Due to shortage of SDLC/SDLI algorithm in middle Doppler targets and expected high complexity in combination structure, the proposed processor is used to overcome this problem beside enhancement of off-pin targets as shown in Fig. 7.

Figure 7 General block diagram of LFMCW radar using the proposed processor.

The integrator of SDLC/SDLI has a stabilization factor, A, of one to ensure the system stability and the proposed filter is used as window function which multiply the incoming signal in time domain with the window function under consideration of same lengths. This multiplication in time domain can be obtained using convolution in frequency domain as in this case which spectral signal is more interest due to using FFT. The coefficients of the proposed filter is chosen to be 1 and −0.5 to solve the problem of middle Doppler frequencies. The proposed filter is chosen a head of first FFT processor which acts as a window function to ensure high detection capability before range-Doppler processor.

The realization of this filter is illustrated as in Fig. 8. For the proposed filter, the difference equation can be written as: (7) y(n)=x(n)−0.5x(n−1)

Where x(n) and y(n) represent the output of FFT processor and the output of the proposed filter respectively. The transfer function of the proposed filter can be written as: Y(Z)=X(Z)(1−0.5Z−1)

Therefore, (8) H(Z)=1−0.5Z−1

The proposed filter is chosen to enhance the detection capability of middle Doppler target velocities which improved using maximization process in Ahmed (2019) with approximately high complexity compared with that of the proposed filter. The simulation of the proposed processor performance and both SDLC/SDLI processor and the traditional algorithm based on MTI only is achieved and discussed in the next section.

Figure 8 Realization of the proposed filter processor.

Computer simulation

Performance of the proposed processor is evaluated using simulation based on Matlab program. The performance is compared with that of both the traditional one and SDLC/SDLI algorithm under the same conditions. It is assumed that, the generation waveform is sawtooth with the central frequency of LFMCW radar (fc) is 24 GHz, bandwidth (B) is 20 MHz, modulation period (Tm) is 80 μsec, number of range cells is 1,024 cells and number of Doppler cells is 32 cells. Comparison between the proposed processor and the traditional one which uses 2D-FFT processor is achieved as shown in Fig. 9. To study the effect of the proposed filter, two scenarios could be applied. First one, for off-pin targets and the other for middle-pin targets. The simulation is performed for these cases under the same conditions to verify a fair comparison.

Figure 9 Block diagram of the proposed processor compared with the traditional 2D-FFT processor.

Off-pin targets

The proposed filter has a great performance on the off-pin target detection. Assume a target in Doppler velocity equals (4.5/15)fm which is off-pin target which lies between Doppler velocities (4/15)fm and (5/15)fm. The target can appear as two targets as in Fig. 10A using the traditional algorithm. But after applying the proposed filter, the target is located at one pin only (at pin number 5) or with Doppler velocity equals (5/15)fm as in Fig. 10B which indicates that, the proposed filter can resolve the problem of off-pin targets and therefore enhance the signal detection.

Figure 10 Response of FFT algorithm.

(A) Before the proposed filter. (B) After the proposed filter.

Middle-pin targets

To evaluate the effect of the proposed filter on the traditional algorithm, a set of moving targets are presented at different Doppler frequencies in noiseless environment (1/32, 4/32, 8/32, 12/32, 16/32, 20/32, 24/32, 28/32, 31/32) × fm. Figure 11 illustrates SDLC/SDLI processor response compared with the traditional algorithm at different Doppler frequencies. It is found that, there are no enhancement in target detection especially for middle-pin targets. Figure 12 represents the response of the proposed algorithm based on the designed filter processor compared with the traditional one at different Doppler frequencies. It is clear that, the proposed processor based on filtering of the signal spectrum has a good performance for both off-pin targets and middle-pin targets compared with both the traditional and SDLC/SDLI processor due to using the maximization selection.

Figure 11 Response of SDLC.

Response of SDLC/SDLI processor compared with the traditional one at different Doppler frequencies.

Figure 12 Response of the proposed processor.

Response of the proposed processor compared with the traditional one at different Doppler frequencies.

Another aspect to evaluate the proposed processor is the detection performance using ROC curve at different Doppler frequencies as shown in Figs. 13 and 14.

Figure 13 ROC of the proposed processor for slow Doppler target.

ROC of the proposed processor compared with that of SDLC/SDLI and the traditional algorithms for slow Doppler target at Pfa of 10−5.

Figure 14 ROC of the proposed processor for middle Doppler target.

ROC of the proposed processor compared with that of SDLC/SDLI and the traditional algorithms for middle Doppler target at Pfa of 10−5.

It is clear that, from Fig. 13, the detection performance of the proposed processor is enhanced compared with both the traditional and SDLC/SDLI processor by nearly 12 dB of SLC/SDLI processor and about 32 dB of the traditional algorithm at slow Doppler target velocity of (2/32)fm. Figure 14 illustrates that, the detection of the target enhanced using the proposed processor by nearly 38 dB of SLC/SDLI processor and about 10 dB of the traditional algorithm at middle Doppler target velocity of (12/32)fm.

Hardware implmentation

The implementation of the proposed processor is very important using FPGA which indicates that it can operate in real-time applications. The implementation is designed for the processing stage which includes; dechirping process of swatooth signal, 2D-FFT processor, proposed filter, MTI, SDLC/SDLI and CFAR detection. Xilinx KC705 DSP kit is used for implementation which includes KINTEX7 XC7K325T FPGA chip which has 241,152 logic cell, 768 DSP slices and about 216 Kbit RAM (Challenges & Solutions, 2012). FPGA board is equipped with an FMC daughter board that contains TI’s ADS62P49/ADS4249 dual-channel 14-bit 250 Msps ADC and TI’sDAC3283 dual channel 16-bit 800 Msps DAC on a daughter board (Abaco Systems, 2013). The FFT core parameters are chosen to be; 32 number of samples, input data width is 32 bits, phase factor width is 24 bits, and Pipelined Streaming, I/O is used. The hardware implementation is performed for both the proposed processor and traditional algorithm which based on SDLC/SDLI. Two targets are simulated at Doppler velocity of pin (2/32) and the other target located at Doppler frequency pin number (12/32) as shown in Figs. 15 and 16. From these figures, it is clear that, the output of the proposed processor can improve the slowly moving target without any effect of other targets. The hardware specifications using Xilinx KC705 DSP kit is summarized in Table 1.

Figure 15 Simulation results of target detection using FPGA.

(A) Traditional algorithm. (B) Proposed processor.

Figure 16 Response of the proposed processor compared with that of both SDLC/SDLI and traditional algorithms using FPGA.

Table 1 FPGA utilization resources of the proposed processor.

Hardware resources	Available resources	Used	Utilization (%)	
Slice registers	326,080	20,697	5	
Slice LUTs	203,800	43,723	21	
RAMB36E1/FIFO36E1s	445	145	32	
RAMB18E1/FIFO18E1s	890	33	3	
DSP48E1s	840	345	40	

The verification of the implementation is performed using Chip scope for the two processors as shown in Fig. 17. It is found that, the Chip scope results met the simulation results as discussed before.

Figure 17 Chip scope result of the proposed processor.

Chip scope result of the proposed processor, SDLC/SDLI and traditional responses.

Conclusion

In this article, detection of targets with small Doppler frequencies has been enhanced using a proposed processor. The enhancement has performed based on filtering process focusing on the detection based on the traditional algorithm using 2D-FFT processor and SDLC/SDLI processor. There are two main problems for target detection with small Doppler frequencies; first one, is the off-pin target detection which traditional algorithm cannot distinguish between these targets. The proposed processor can resolve this problem. Second problem, is the detection of middle-pin targets which is the main problem for SDLC/SDLI processor and this case has been overcame using maximization process but it suffer from high complexity. So, this problem can be resolved using the proposed algorithm based on a proposed filter as a head of the first FFT processor with less complexity compared with maximization process.

The performance of the proposed processor is examined compared with that of the traditional one and SDLC/SDLI processor through these two points. The detection performance of these targets can be evaluated using ROC curves at different target velocities and at low probability of false alarm.

It is found that, the detection performance of the proposed processor is enhanced by nearly 12 dB of SLC/SDLI processor and about nearly 32 dB of the traditional algorithm at slow Doppler target velocity and about nearly 38 dB of SLC/SDLI processor and 10 dB of the traditional algorithm at middle-Doppler target velocity. The implementation of the proposed processor is achieved using FPGA and Chip scope. It is found that, it meets the simulation results.

Supplemental Information

Supplemental Information 1 Matlab programs.

Click here for additional data file.

Additional Information and Declarations

Competing Interests

Author Contributions

Data Availability

The authors declare that they have no competing interests.

Sameh Ghanem conceived and designed the experiments, performed the experiments, analyzed the data, performed the computation work, prepared figures and/or tables, authored or reviewed drafts of the paper, and approved the final draft.

The following information was supplied regarding data availability:

Code is available in the Supplemental Files.

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
