# Peer review of "Enhancement of small doppler frequencies detection for LFMCW radar"

_PeerJ Computer Science, doi:10.7717/peerj-cs.367_

## Round 0.1 · original submission · Major Revisions

Major revision. The paper has contributed toward the research idea.

Reviewer 1 ·

Basic reporting

The English of the paper is better, and the charts basically meet the requirements.

Experimental design

This paper studies the problem of slow target detection, which has good significance.

Validity of the findings

The method proposed in this paper is novel , but the advantages of this method are not described clearly. There is a lack of in-depth theoretical and experimental research and verification, and lack of comparison with other methods.

Additional comments

The problems studied in this paper have good practical significance. However, the method studied in this paper lacks in-depth theoretical and experimental verification, and the comparison between the proposed method and other methods is also insufficient.The suggestions are as follows::
1. The whole theoretical process of the research method is given.
2. In this paper, the target velocity range of the proposed method should be given, and explicit simulation and practical verification examples should be given.
3. In this paper, the relationship between detection method and target SINR should be given.

Reviewer 2 ·

Basic reporting

1- The purpose of the paper is ambiguous and not clear relative to the mentioned
references and previously published work :-
- Reference no. 6
- "Detection improvement of off-pin targets in FMCW radars", October 2019IOP
Conference Series Materials Science and Engineering 610:012048, DOI:
10.1088/1757-899X/610/1/012048. Not included in the references.
2- For the proposed filter:
- The reason of choosing its coefficients and order is not clear, and needs more proof.
- The reason for putting it before the first FFT or the second FFT needs explanation.
- References [7] and [8] are not found in the paper body.

Experimental design

1- Research questions are not well defined.
2- Results of Fig (10) is the same as that proposed in [6] . No extra enhancement.
3- According to Fig (10), the mentioned enhancement in Fig. (11) has no meaning.
4- Hardware design and implementation was not described and its results are not sufficient.

Validity of the findings

1- No novality was found in this article.
2- Results in conclusion need to be verified.

---

## Round 0.2 · accepted · Accept

All comments have been met by the author. I am happy with their comments and modifications.